# Association between cardiac autonomic regulation, visceral adipose tissue, cardiorespiratory fitness and ambient air pollution: 4HAIE study (Program–4)

**Tomas Dostal**[1]*, **Martina Dankova**[2], **Tomas Barot**[3], **Dominik Sindler**[1], **Petr Kutac**[1], **Vera Jandackova**[1], **Daniel Jandacka**[1], **Peter Hofmann**[4], **Lukas Cipryan**[1]

**1** Department of Human Movement Studies & Human Motion Diagnostic Centre, University of Ostrava, Ostrava, Czech Republic, **2** Institute for Research and Applications of Fuzzy Modelling, University of Ostrava, CE IT4Innovations, Ostrava, Czech Republic, **3** Department of Mathematics with Didactics, University of Ostrava, Ostrava, Czech Republic, **4** Institute of Human Movement Science, Sport & Health, Exercise Physiology, Training & Training Therapy Research Group, University of Graz, Graz, Austria

* dostaltomas.ov@gmail.com

**Data Availability Statement:** The explicit and specific instructions for requesting data are outlined in the document The Order of the Dean of

## Abstract

### Background

The main objective of the present cross-sectional cohort study was to determine whether there is an association between cardiac autonomic regulation, as expressed through heart rate variability (HRV), and cardiorespiratory fitness (CRF), visceral adipose tissue (VAT), and over the long-term living in areas with low or high air pollution.

### Methods

The study sample included 1036 (487 females) healthy runners (603) and inactive participants (age 18–65 years) who had lived for at least 5 years in an area with high (Moravian-Silesian; MS) or low (South Bohemian; SB) air pollution in the Czech Republic. A multivariable regression analysis was used to evaluate the associations between multiple independent variables (CRF (peak oxygen consumption), VAT, sex, socioeconomic status (education level), and region (MS region vs. SB region) with dependent variable HRV. The root mean square of successive RR interval differences (rMSSD) was employed for the evaluation of HRV.

### Results

The multivariable linear regression model revealed that cardiac autonomic regulation (rMSSD) was significantly associated with CRF level (p < .001) and age (p < .001). There were no associations between rMSSD and region (high or low air-pollution), sex, education level or VAT (p > 0.050).

FoE No. 00/2020: Rules Regarding Project 4HAIE
Data Access and Usage, which is freely available in
the "Data Access" section of the HAIE website
(https://www.4haie.cz/en/data-2/). The specific
variables included in the final analysis of this study
are listed in the data selector section under the
"Data Access" section as follows: General ID
(general ID)—age (18 to 65 years), gender (males
and females), and region (MSR and SB); Functional
Anthropology (DXA datasets: whole body)—
vfat_area; Exercise Physiology (HRV datasets)—
s1_rmssdms; Exercise Physiology (GXT)—
2vo2max2; Behavioral Lab (SES questionnaire; A
General Part; A7–A12: Education and work activity)
—a7.

**Funding:** This work has been produced with the
financial support of the European Union under the
LERCO project (CZ.10.03.01/00/22_003/0000003)
via the Operational Programme Just Transition and
from the project Research of Excellence on Digital
Technologies and Wellbeing (CZ.02.01.01/00/
22_008/0004583) which is co-financed by the
European Union. The funders had no role in study
design, data collection and analysis, decision to
publish, or preparation of the manuscript.

**Competing interests:** The authors have declared
that no competing interests exist.

## Conclusions

We showed that living in an area with low or high air pollution is not associated with cardiac autonomic modulation in healthy runners and inactive individuals. CRF and age significantly directly and inversely, respectively, associated with HRV. There were no other significant associations.

## Introduction

The incidence and prevalence of non-communicable diseases such as cardiovascular disease (CVD), obesity, metabolic syndrome, and type 2 diabetes mellitus has increased rapidly over the last two decades, resulting in it being recognized as a global crisis [1]. A low level of heart rate variability (HRV) is one of the various risk factors for chronic diseases [2–4]. The established, non-invasive method of HRV evaluates cardiac autonomic regulation and individual risk for disease or lifestyle changes [4]. A meta-analysis of cohort studies by Fang et al.[3] showed that, for individuals with CVD, those with reduced HRV had a 112% higher risk of all-cause mortality and a 46% higher risk of incurring a cardiovascular event compared to those with high HRV. Furthermore, Gidron et al. [5] highlighted the potential usefulness of measuring vagal nerve activity via HRV as a method of detecting other non-communicable diseases (not just CVD) such as metabolic syndrome, cancer, insulin resistance, and chronic obstructive pulmonary disease. Increasing vagal nerve activity could have also have a beneficial effect on executive functions that include: self-regulation, inhibition, memory, and problem solving in patients with neurodegenerative conditions [6]. A similar pattern was shown by Wulsin et al. [7] who indicated HRV was a significant predictor of hyperglycaemia and high blood pressure within a 12-year period during The FHS Offspring Cohort Study. Further, it was shown that HRV predicted the development of CVD and diabetes type II in the male study sample [7].

Cardiorespiratory fitness (CRF) level is defined as a component of physiological fitness related to the ability of the circulatory and respiratory system to supply oxygen during sustained physical activity [8]. There is sufficient scientific evidence supporting that a low CRF level is a significant risk factor for all-cause mortality, morbidity and cardiovascular diseases. For example, a dose-response analysis performed by Han et al. [9] showed that increasing CRF by one MET was associated with a 12%, 13% and 7% reduced risk for all-cause, CVD and cancer mortality, respectively. In addition, results from the UK Biobank cohort study showed that a higher level of CRF was associated with lower incidence of CVD, respiratory diseases and colorectal cancer [10]. Indeed, each MET higher fitness was associated with lower risk for all-cause mortality, circulatory disease mortality and respiratory disease mortality [10]. Along with CRF, overweight and obesity are well-known related risk factors. Windham et al. [11] showed that central adiposity, measured via waist circumference (WC), was associated with lower HRV. However, the measurement of WC itself is highly affected by the experience of the researcher and correlation with visceral adipose tissue (VAT) in the literature presented a wide range [12]. These results are in line with the CARDIA study, which either showed WC was a strong determinant of HRV. However, these strong associations between HRV and adiposity in the CARDIA study were attenuated after CRF variables were included in the analysis, which proved to be a stronger predictor of HRV.

Another potential risk factor that may influence HRV is air pollution. An analysis of the 2015 Global Burden of Diseases study revealed that ambient particulate matter pollution was

the fifth highest risk factor for deaths after high systolic blood pressure, smoking, high fasting plasma glucose and high total cholesterol [13]. Moreover, ambient air-pollution substantially contributed to 17% of ischemic heart disease, 14% of cerebrovascular disease, 16% of lung cancer, 25% of lower respiratory infection, and 27% of chronic obstructive pulmonary disease [13]. Findings of an umbrella review of systematic reviews and meta-analyses, from studies conducted mainly in low and middle income countries, showed that short-term exposure to air pollutants $PM_{2.5}$, $PM_{10}$, and $NO_x$ was consistently associated with myocardial infarction (MI), hypertension, and stroke mortality and morbidity [14]. Furthermore, long term-exposure to air-pollutant $PM_{2.5}$ was associated with increased incidence of MI, atherosclerosis, hypertension, and stroke mortality and morbidity [14]. The ambient air quality in the Czech Republic has significantly improved over the past 70 years, but regional differences persist. The Moravian-Silesian region remains one of the most polluted areas in Europe, with high levels of particulate matter (PM) and benzo[a]pyrene (BaP) due to heavy industry [15].

The aim of this study was therefore, to determine whether there is an association between cardiac autonomic regulation, visceral adipose tissue (VAT), cardiorespiratory fitness (CRF) and long-term exposure to high or low air-pollution adjusted for age, sex and educational level in healthy runners compared to inactive individuals.

## Methods

### Participants

The main purpose of the prospective, longitudinal and multidisciplinary cohort 4HAIE study was to evaluate the impact of living in an air-polluted environment on physiological, biomechanical, psychosocial and anthropometric variables in male and female runners and inactive controls. A total of 1314 participants were recruited from two different regions within the Czech Republic that differ in air quality [16]. The highly polluted Moravian-Silesian Region (MSR, N = 750) was compared with South Bohemia (SB, N = 564) as the control region with low air-pollution. A two-day laboratory assessment was performed by all participants including a series of physiological, biomechanical, anthropometric, cognitive and MRI assessments, which has been described in more detailed in the protocol papers [17–19]. To be included in the study, participants had to meet the following criteria: non-smoker, aged between 18–65 years at the start of the study. Runners had to meet public PA guidelines (150 min/week moderate or 75 min/week vigorous PA or an equivalent combination of the two as per WHO recommendations; be running at least 10 km per week in the last 6 weeks and plan to continue running for the next 12 months). Inactive controls did not meet the public PA recommendations but were capable of normal PA including running. Both runners and inactive participants were required to have lived in the recognized regions for at least 5 years with the prerequisite not to move residency to another region in the next 12 months. Exclusion criteria included: being a smoker, having an acute health problem (pain /injury / surgery) within the last 6 weeks preventing from physical activity, having contraindications to magnetic resonance imaging or dual-energy x-ray absorptiometry (DXA) [19].

The present study includes data from 1036 participants from the original 4HAIE data set. Only data from participants who had no missing data in the variables analysed and no detected outliers as observation via Cook´s distance (see details in statistical section) were included in the analysis. Based on this, 278 participants were excluded from the analyses. The 4HAIE study is conducted in accordance with the Declaration of Helsinki. The study was approved by the Ethics Committee of the University of Ostrava (3/2018). Participant were provided detailed information sheets and a written informed consent was obtained from all participants.

### High air-polluted region versus low-polluted region

The Czech Republic follows the European Union (EU) (EU air quality standards (europa.eu)) for PM10, PM2.5, NO2, benzene, and BaP limit values, which are 40 µg/m3, 20 µg/m3, 40 µg/m3, 5 µg/m3, and 1 ng/m3, respectively. Moravian-Silesian region (MSR) defined by six districts is located at the eastern part of the Czech Republic and is among the highest air-polluted areas from fourteen high-polluted European Union regions. On the other hand, the South Bohemia region (SBR) located at the opposite part of the Czech Republic and defined by seven districts is a resource-poor area without significant sources of raw energy materials compared to MSR and is characterized as low-polluted region [20]. Long-term differences in air pollution between the MSR and the SBR published by Machaczka et al.[16] for the time period 2000–2017 showed that these EU standards were exceeded in the MSR region on average for PM10 by 12.3%, for PM2.5 by 75%, and for BaP by 160%, without exceeding of the limit values in the SB region (all data are shown in S1 Table). The cross-sectional data collection used for this study ran from 1. April 2019 to 30. August 2021. Based on data from the Czech Hydrometeorological Institute, **Fig 1A and 1B** show the differences in the five-year (2017–2021) average of annual mean concentrations between regions for the air pollutants PM2.5 and BaP [21].

### Cardiac autonomic regulation

Cardiac autonomic regulation was assessed by heart rate variability (HRV) measures and for the purpose of this study HRV was expressed by the time-domain most robust vagal modulation variable root mean square value of successive differences of the normal RR intervals (rMSSD) [22]. 10-min of recording in supine position was performed after waking in the morning (until 7:00 am) before breakfast in a quiet room without any external disturbances at the start of the second-day laboratory assessment. Participants were also instructed the evening before to remain in bed upon waking up and limit their activities to only essential tasks, such as using the bathroom, until the researcher arrived to carry out the HRV measurement. The first 5 minutes of the recording were used to achieve steady-state conditions and this part was disregarded for the analysis. ECG data obtained from a Bittium Faros device (Bittium Inc, Oulu, Finland) from two electrodes placed on a chest belt were then imported into Kubios

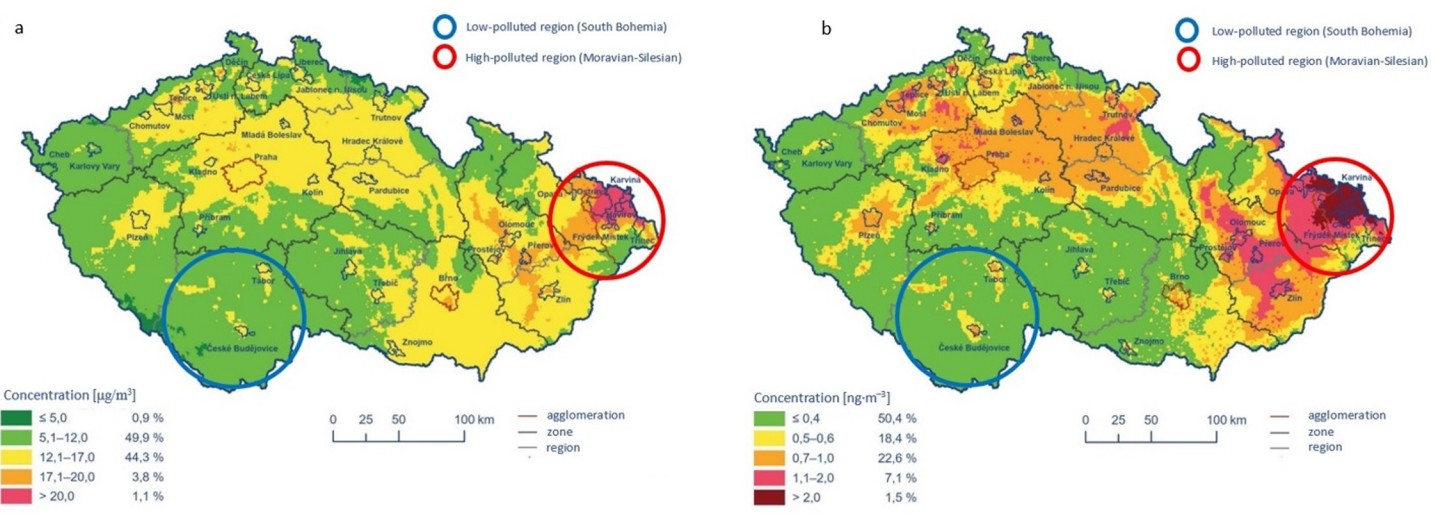

**Fig 1. a-b.** Five years (2017–2021) average of annual mean concentrations of the PM 2.5 (1a) and benzo(a)pyren (CHMU, 2021).

HRV Premium software (Kubios Oy, Kuopio, Finland). The length of the analysed record was 5 minutes according to task force recommendations [23]. Before extracting the R-wave time points, the R-waves were interpolated to 2000 Hz. Analysis began with visual inspection for artefacts and through an automated artefact detection algorithm, which was also applied by the Kubios software. All visual inspections and edits were consulted and agreed by the team of 3 researchers. Artefacts included missing, redundant, or misaligned rhythm detections, as well as ectopic rhythms such as premature ventricular contractions or other arrhythmias which were filtered out from the RR interval time series used for HRV analysis. Slow nonstationary oscillations in the RR intervals were detrended using smoothness priors [17].

## Graded exercise test (GXT)

As a measure of CRF level, maximum oxygen uptake ($VO_{2peak}$) was obtained from a GXT test. The protocol of the GXT started with 3 min of walking at 5.0 km/h to familiarise subjects with the treadmill. The GXT protocol then commenced at 6.0 km/h, with speed subsequently increasing by 1.0 km/h every minute until volitional exhaustion at an inclination remaining at 1% throughout the test. Expired air was continuously monitored to analyse $O_2$ and $CO_2$ concentrations during the GXT with a breath-by-breath system (Blue Cherry, Geratherm Medical AG, Germany) which was calibrated according to the manufacturer's guidelines. The highest average $O_2$ consumption measured over a 30 s period prior to completing the GXT was used to determine $VO_{2peak}$. More detailed information regarding exercise performance have been presented recently [24]. Participants who did not pass the Physical Activity Readiness Questionnaire [25], were not allowed to perform the GXT unless explicit permission (given by a medical doctor) was provided. Blood pressure (BP) was also checked before the GXT. In case of BP values exceeding 150/90 mm Hg, participants were not allowed to perform the GXT, but continued in the study protocol. Each participant was advised not to participate in any vigorous activity 24 hours prior to the test [17].

## Anthropometry

Dual-energy X-ray absorptiometry (DXA) was used as the gold standard method for estimating visceral adipose tissue (VAT). The measurements were executed using the Hologic Horizon A bone densitometer (Hologic Discovery A, Waltham, Massachusetts, USA). For the DXA measurements, the height (InBody 370 stadiometer, Biospace, South Korea) and weight (InBody 770 bioimpedance, BioSpace, South Korea) of the participants were used. A whole-body scan was used for the analysis of body composition, including segmental analyses [17].

## Statistical analysis

A linear regression model with multiple independent variables and one dependent variable was employed. The dependent variable in the model was the heart rate variability variable rMSSD. Independent variables included CRF level expressed as maximal oxygen uptake ($\dot{V}O_{2peak}$), visceral adipose tissue (VAT) and age, sex, region of residency, and socioeconomic status represented by education level were used as covariates. For the analysis, the level of education was divided into the three following categories: Education level 1 (basic, apprentice, secondary vocational without GSCE), Education level 2 (secondary school diploma–general and vocational), Education level 3 (higher vocational–post secondary, university and higher). All numerical data were controlled for normal distribution using the Shapiro-Wilk test, which yielded negative results. A logarithmic transformation (log10) was applied to achieve normal distribution at a significance level of 0.01 for rMSSD and VAT (data before transformation is given in S2 Table). Additionally, the residuals vs. fitted plot was utilized to assess the

assumptions of linearity and homoscedasticity in the linear model. Cook's distance was used to identify influential observations within the model. Observations with Cook's distance exceeding the value two have a greater potential to impact the estimated coefficients of the model as well as homoscedasticity were excluded from further analyses. The provided summary includes information about model coefficients, standard errors, t-statistics, p-values, and other statistical indicators at a significance level of 0.05.

## Results

254 participants were excluded due to missing data after initial data screening. Further, after the first phase of data analyses, an additional 24 participants were excluded because they were identified as influential observations within the model (Cook's distance). The total number of participants included in the final analysis model was 1036 and their basic descriptive statistics are shown in **Table 1**.

The multivariable linear regression model revealed that rMSSD was significantly and directly associated with CRF level ($p < .001$) and significantly but inversely associated with age ($p < .001$). There was no effect of location (high or low air-polluted region), sex, education level or VAT on rMSSD data ($p > .050$). The results of the multivariable linear regression model are displayed in **Table 2**.

Significant associations between rMSSD and CRF level are displayed in **Fig 2A–2C** (distributed by education 2a; region 2b; sex 2c). Significant associations between rMSSD and age are displayed in **Fig 3A–3C** (distributed by education 3a; region 3b; sex 3c).

## Discussion

We investigated the association between cardiac autonomic regulation, VAT, CRF and long-term living in areas with historically low and high air pollution in the Czech Republic in healthy runners and inactive individuals. The multivariable linear regression model revealed that cardiac autonomic regulation (as measured by rMSSD) was significantly associated with CRF level and age. There were no associations between cardiac autonomic regulation and living in high or low air-pollution regions, sex, education level or VAT.

The impact of age on cardiac autonomic regulation is well established. Our results are in line with Fluckiger et al. [6] who examined the impact of age on and blood pressure (BP) variability in 65 healthy participants aged 47±18 years. It was indicated that time-domains

**Table 1. Characteristics of the participants.**

| | |
|---|---|
| Participants (female/male) | 1036 (487/549) |
| Moravian-Silesian Region/ South Bohemia Region | 569/467 (55%/45%) |
| Age | 38.1 (12.5) |
| Height (cm) | 174.6 (9.0) |
| Body Mass (kg) | 75.2 (14.3) |
| rMSSD (ms) | 1.58 (.28) [#] |
| $V$O$_{2peak}$ (ml/min/kg) | 41.6 (10.3) |
| VAT$_{area}$ (cm$^2$) | 1.83 (.24)[#] |
| Education level 1 *(basic, unfinished or apprentice)* | 135 (14%) |
| Education level 2 *(secondary school diploma)* | 420 (40%) |
| Education level 3 *(higher vocational, university and higher)* | 481 (46%) |

VO$_{2max}$–maximal aerobic power, rMSSD - the root mean square value of the successive differences of the normal RR intervals, VAT–visceral adipose tissue. Data are expressed as mean ± SD, [#] - logarithmic data.

**Table 2. Results of the multivariate linear regression model.**

| ln rMSSD: model $R^2$ = 0.207; Residual Standard Error (RSE) = 0.25; F-statistic = 44.87; P = < 0 | | | | |
|---|---|---|---|---|
| **Variable** | **Regression coefficient** | **Standard Error** | **t value** | **P value** |
| Location | -0.0009 | 0.0147 | -0.065 | .948 |
| Sex | -0.0402 | 0.0246 | -1.636 | .102 |
| *Age* | *-0.0091* | *0.0007* | *-12.432* | *< .001* |
| Education level 2 | 0.0091 | 0.0233 | 0.393 | .694 |
| Education level 3 | -0.0302 | 0.0233 | -1.297 | .195 |
| VAT$_{area}$ (cm$^{2)}$) | -0.0056 | 0.0518 | -0.110 | .913 |
| *VO$_{2peak}$ (ml/min/kg)* | *0.0066* | *0.0011* | *5.699* | *< .001* |

VO$_{2peak}$−peak oxygen consumption, rMSSD - the root mean square value of the successive differences of the normal RR intervals, VAT–visceral adipose tissue.

normalized high and low-frequency HRV was negatively correlated with age [26]. Similarly, Antelmi et al. [27] showed a reduction in monitored time and frequency domain variables with age from 24-hour electrocardiographic recordings in 653 participants. Lastly, the analysis of 10 years of data from the population-based longitudinal Whitehall II cohort study in the UK showed decreasing trends for HRV variables with increasing age [28].

## Cardiac autonomic regulation and cardiorespiratory fitness

We found significant association between HRV and CRF (expressed as maximal oxygen uptake; $\dot{V}O_{2peak}$). This finding is in line with Medeiros et al. [29] who also found a significant relationship between CRF assessment (by the Leger shuttle run test) and cardiac autonomic regulation (expressed by rMSSD). Similarly, Kiviniemi et al. [30] revealed a significant correlation between CRF and cardiac autonomic regulation (assessed by rMSSD variables) in 3144 participants (1761 females). Our results and results above indicate that improving the CRF level could be one of the best options for achieving improvements in cardiac autonomic modulation. Cardiorespiratory fitness is partly genetically determined, but can be improved substantially by regular physical activity or exercise [31]. The Danish Health Examination Survey study by Eriksen et al. [32] of 16 025 participants aged 18–65 years (DANHES) showed that CRF level, obtained from a GXT and expressed as VO$_{2max}$, was inversely associated with total daily sitting time. The harmful effects of a lack of leisure time physical activity (LTPA) on CRF were presented in the results of the National Health and Nutrition Examination Survey (NHANES) in US adults aged 20–49 years [33]. Furthermore, the DANHES study, observed that participants who reported lower amounts of LTPA also had lower CRF levels and *vice*

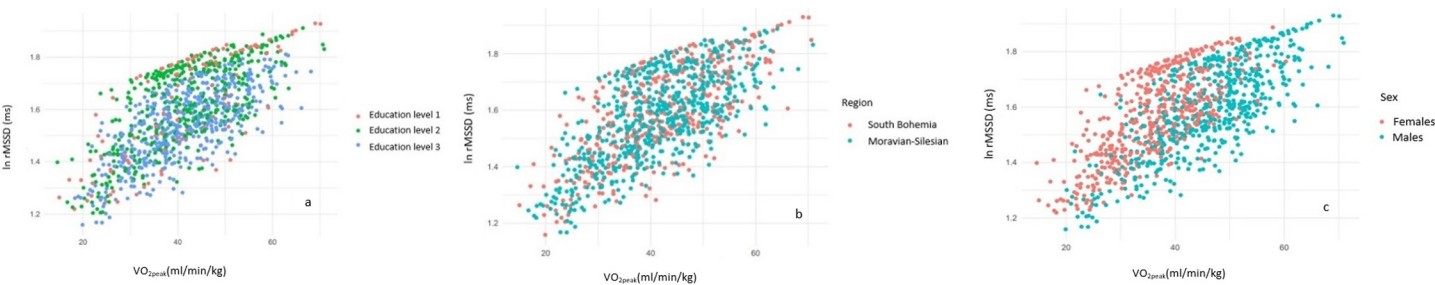

**Fig 2. a-c.** Associations between heart rate variability and CRF level expressed as $\dot{V}O_{2peak}$ with education, region and sex as confounding factors. CRF–cardiorespiratory fitness.

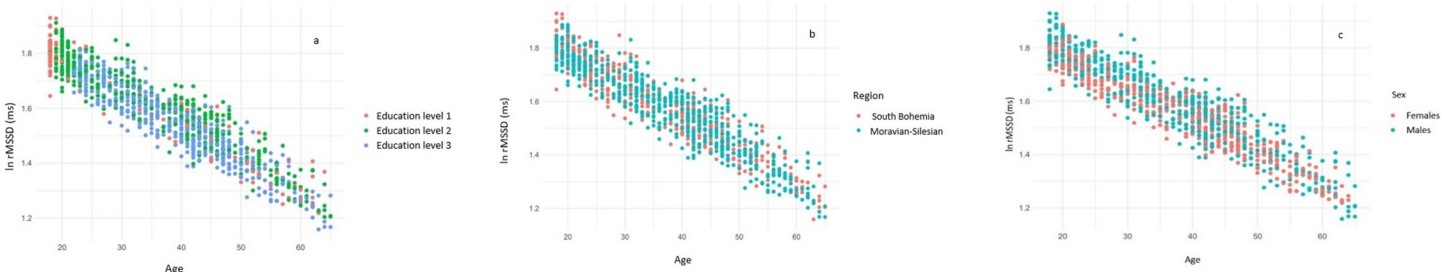

**Fig 3. a-c.** Associations between heart rate variability and age with education, region and sex as confounding factors.

*versa*. Lastly, the Finish population-based study MOPO also indicated a positive association between the amount of the physical activity and cardiac autonomic regulation in 3395 adolescent males [34]. The individual CRF level was estimated indirectly via the Polar Fitness Test$^{TM}$ which is based on resting heart rate, heart rate variability, gender, age, body weight, height and the self-assessment of the level of long-term physical activity. Participants who reported "high" or "top" amounts of PA/week had significant higher rMSSD compared to the "low" and "moderate" groups.

Although we did not take the intensity of the physical activity into account in our analysis, Kim et al.[35] showed that exercise intensity could have beneficial effects on cardiac autonomic regulation. It was observed that high-intensity interval training significantly increased CRF levels and rMSSD compared to the control/no exercise group in 34 male regular smokers [35]. However, no significant differences were observed between the moderate training group and control/no exercise group for either CRF or rMSSD [35] highlighting the importance of exercise intensity.

## Cardiac autonomic regulation and long-term exposure to air-pollution

Our analyses did not find any association between HRV and residing for at least 5-years in a high or low-polluted region within the Czech Republic. These results are contrary to the results of other studies evaluating short-term exposure to ambient air-pollution. For example, the population-based ARIC study by Liao et al. [36] demonstrated significant adverse effects of higher levels of $PM_{10}$, ozone, carbon monoxide, nitrogen dioxide, and sulphur dioxide on cardiac autonomic control, however did not control for CRF. Similar acute, negative trends were observed for HRV after exposure to low-levels of $PM_{2.5}$ in a small sample of participants (n = 10) [37].

The potential positive acute effects of physical activity on cardiac autonomic function in a polluted environment have been observed previously [38]. Cole Hunter et al. [38] examined the acute effects of exercising in low and high-traffic environments on cardiac autonomic function. The authors found that physical activity performed even in high traffic-induced air pollution environments still had a beneficial acute effect on 24-hour cardiac autonomic regulation. It was speculated that physical activity may offset the impact of high air pollution on parasympathetic modulation of the heart even at higher levels of air pollution exposure [38]. However, the CRF level of the participants was not mentioned, and so it is unclear whether being physically active or also having a certain, baseline level of fitness is the cause for such beneficial effects [38]. Definitely, there is a lack of long-term exposure to ambient air-pollution studies. One rare study evaluated the short-term and 1-year long effect of the exposure to $PM_{2.5}$ and black carbon (BC) particulate air-pollution in an elderly cohort of the Normative Aging Study [39]. This study included result from 540 males and HRV time and frequency

domain variables were obtained from 12 min recordings in a sitting position where the last 7 min were used for final analyses. The results of this study confirmed adverse effects of a short-term exposure to air-pollution on heart rate variability. On the other hand, evaluation of 1-year long exposure to $PM_{2.5}$ and BC showed similar patterns which did not provide any significant strong association between pollutant and heart rate variability and provided inconsistent results [39]. It also needs to be critically noted that CRF was not identified as an important confounding factor in this study [39]. Therefore, the long-term exposure to air pollution or the controlling of CRF may explain why no association was found with HRV and air pollution in the current study. Moreover, the discrepancy between our findings and studies reporting a negative impact of air pollution on heart rate variability could be also attributed to methodological differences. In particular, our approach of dividing participants into experimental and control groups solely on the basis of regional pollution differences is relatively new in this field. This methodological distinction may contribute to the different results compared to studies that directly measured exposure to pollutants or quantified individual exposure levels. In addition, our study cohort, which included healthy runners and inactive individuals aged 18–65 years, stratified by sex and age, differs from other studies that predominantly included smaller samples, individuals at increased cardiovascular risk or with pre-existing cardiovascular disease, or older adults.

Based on scientific evidence, we know that physical activity has a positive effect to increase an individual's cardiorespiratory fitness, which has protective effects against the development of cardiovascular or other non-communicable diseases, as well as a positive effect on cardiac autonomic regulation. On the other hand, due to the lack of research investigating the engagement in physical activity within polluted environments there is currently no clear consensus on whether physical activity in an air polluted environment has a clear beneficial or adverse impact on an individual's health. One such investigation, a systematic review conducted by Gandhi et al. [40] demonstrated that the cardiovascular benefits of outdoor physical activity can outweigh the adverse effects of air pollution. However, the authors also noted that this positive effect diminishes when physical activity is performed in environments with high levels of pollutants, particularly fine particulate matter ($PM_{2.5}$) [40]. Similarly, Andersen et al. [41] explored the combined effects of physical activity and air pollution on mortality among elderly urban residents through data from the Danish Diet, Cancer, and Health cohort. Their findings indicated that physical activity, even in areas with elevated ambient nitrogen dioxide ($NO_2$) concentrations, may attenuate, but does not entirely negate its beneficial effects on respiratory mortality [41]. Due to the increasing prevalence of urbanization, it is evident that further longitudinal studies are required to definitively determine whether engaging in physical activity within polluted air exerts a beneficial or detrimental impact on one´s health. Our results however, in general do not show an increased risk of being active in polluted regions at least for the investigated healthy group of participants, a finding which was independent from age and sex.

### Cardiac autonomic regulation and abdominal obesity

Sympathetic nerve stimulation has been shown to increase fatty acid release, and either a decrease in sympathetic activity or an increase in parasympathetic activity was causally related to an accumulation of white adipose tissue [42]. However, in this specific cohort of participants, we did not find an association between cardiac autonomic regulation and VAT, which is in contrast to Windham et al. [11] who found a significant association between WC and cardiac autonomic regulation expressed as rMSSD [11]. Differences between our results and Windham et al. [11] could be explained by the different variables measured (waist circumference vs DXA) for expressing VAT. Research assessing the relationship between cardiac

autonomic regulation and the amount of VAT is lacking. Some rare studies with inconsistent results only use the rather weak parameter BMI to express the degree of obesity.

## Strength and limitations

The primary strengths of this study are the large cohort stratified for sex, age and region, which were clearly defined by differences in air pollution according to European Union standards. The study also reveals that CRF is a significant predictor of cardiac autonomic modulation, influenced by activity and age, but not by living in an area with high air pollution over the long-term.

However, there are several limitations, which have to be considered. First, we acknowledge that the cross-sectional design does not allow to us assess and discuss causalities which could be addressed by the forthcoming second wave of measurement. Second, we did not obtain direct information on lifetime exposure to air pollution on an individual level and actual air-pollution exposure may vary throughout the year and within regions. Third, the study included only healthy participants which does not allow to evaluate valuable effects of air pollution on at-risk subjects and to generalize the results to the overall population.

## Practical application

Our study results showed no association between living in areas with high or low pollution and HRV. On the other hand, we found a significant and strong association between the CRF level and HRV. These findings point to the importance of a high CRF level (e.g., through increased physical activity and exercise training) to achieve improvements in cardiac autonomic regulation and other health related risk factors regardless of living in low or high air polluted regions. It is therefore, recommended to include CRF measures in all future studies as CRF is likely a strong confounder, which has been rarely included in most studies regarding health concerns.

## Conclusions

The present study shows that cardiorespiratory fitness and age were significantly associated with heart rate variability independent of living in a high air-polluted region and the amount of visceral adipose tissue in healthy runners and inactive individuals. We conclude that cardiac autonomic activity is related to cardiorespiratory fitness but not to living in areas with high air pollution. The benefits of regular physical fitness may outweigh the detrimental effects of living in high AP areas but more studies with detailed assessment of exposure to specific air pollutants are needed.

## Supporting information

**S1 Table. Differences between MSR and SBR as long-term averages of the exposure to air pollutants in 2000–2017 [16].** $VO_{2peak}$–maximal aerobic power, rMSSD - the root mean square value of the successive differences of the normal RR intervals, VAT–visceral adipose tissue. Data are expressed as mean ± SD (or *median ±IQR).*
(DOCX)

**S2 Table. Characteristics of the participants (data before logarithmisation).** MSR–Moravian-Silesian Region, SBR–South Bohemia Region, EU–European Union, PM–Particulate matter, $NO_2$ –nitrogen dioxide. Data are expressed as mean ± SD.
(DOCX)

## Acknowledgments

We extend our gratitude to Matthew Zimmermann, Ph.D., for his valuable assistance in proof-reading this manuscript.

## Author Contributions

**Conceptualization:** Tomas Dostal, Vera Jandackova, Daniel Jandacka, Peter Hofmann, Lukas Cipryan.

**Data curation:** Petr Kutac, Vera Jandackova, Daniel Jandacka, Lukas Cipryan.

**Formal analysis:** Tomas Dostal, Martina Dankova, Tomas Barot.

**Funding acquisition:** Daniel Jandacka.

**Investigation:** Tomas Dostal, Dominik Sindler.

**Methodology:** Petr Kutac, Vera Jandackova, Daniel Jandacka, Peter Hofmann, Lukas Cipryan.

**Supervision:** Petr Kutac, Vera Jandackova, Daniel Jandacka, Lukas Cipryan.

**Validation:** Tomas Dostal, Daniel Jandacka, Lukas Cipryan.

**Visualization:** Tomas Dostal.

**Writing – original draft:** Tomas Dostal, Peter Hofmann, Lukas Cipryan.

**Writing – review & editing:** Tomas Dostal, Martina Dankova, Tomas Barot, Dominik Sindler, Petr Kutac, Vera Jandackova, Daniel Jandacka, Peter Hofmann, Lukas Cipryan.

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
