## [Decision Letter · Decision Letter 0]

17 Oct 2024

PONE-D-24-39037Association between cardiac autonomic regulation, visceral adipose tissue, cardiorespiratory fitness and ambient air pollution: 4HAIE Study (Program – 4).PLOS ONE

Dear Dr. Dostal,

Thank you for submitting your manuscript to PLOS ONE. After careful consideration, we feel that it has merit but does not fully meet PLOS ONE’s publication criteria as it currently stands. Therefore, we invite you to submit a revised version of the manuscript that addresses the points raised during the review process.

We look forward to receiving your revised manuscript.

Kind regards,

Aida Fallahzadeh

Guest Editor

PLOS ONE

Journal Requirements:

“The baseline data refers to the project funded by the Czech Ministry of Education, Youth and Sports, the project 4HAIE “Healthy Aging in the Industrial Environment - Program 4” (CZ.02.1.01/0.0/0.0/16_019/0000798) within its sustainability period.”

“This work has been produced with the financial support of the European Union under the LERCO project (CZ.10.03.01/00/22_003/0000003) via the Operational Programme Just Transition and from the project Research of Excellence on Digital Technologies and Wellbeing (CZ.02.01.01/00/22_008/0004583) which is co-financed by the European Union.”

“This work has been produced with the financial support of the European Union under the LERCO project (CZ.10.03.01/00/22_003/0000003) via the Operational Programme Just Transition and from the project Research of Excellence on Digital Technologies and Wellbeing (CZ.02.01.01/00/22_008/0004583) which is co-financed by the European Union.”

Please include this amended Role of Funder statement in your cover letter; we will change the online submission form on your behalf

4. Please note that your Data Availability Statement is currently missing a direct link to access each database. If your manuscript is accepted for publication, you will be asked to provide these details on a very short timeline. We therefore suggest that you provide this information now, though we will not hold up the peer review process if you are unable.

Additional Editor Comments :

Dear Dr. Dostal,

Thank you for your submission of the manuscript titled “Association between cardiac autonomic regulation, visceral adipose tissue, cardiorespiratory fitness and ambient air pollution: 4HAIE Study (Program – 4)” to our journal. After careful consideration, I am pleased to inform you that your manuscript has been provisionally accepted, pending minor revisions.

While the manuscript is well-prepared and contributes significantly to the field, I believe that addressing the reviewer's points will enhance its quality.

Please revise your manuscript accordingly and provide a detailed response to each point. I look forward to receiving your revised manuscript by September 15th,2024.

Thank you for your valuable contributions to our journal.

Best regards,

Aida Fallahzadeh, MD

Reviewer 1:

Methodological Considerations:

1. What methodology was employed to assess individual physical activity levels? Was self-reporting utilized? How was strength training accounted for in the analysis?

2. Were ambient air pollution levels measured concurrently with HRV assessments? If not, how was temporal variability in air quality addressed?

3. The study design appears to have omitted consideration of physical activity intensity. What was the rationale for this exclusion, and how might it impact the interpretation of results?

4. The broad exclusion criteria may introduce selection bias. What measures were implemented to mitigate this potential confounding factor?

Results Interpretation and Discussion:

5. The discrepancy between this study's findings and previous literature warrants further elucidation. A more comprehensive comparative analysis of methodologies and populations might shed light on these inconsistencies.

6. The manuscript would benefit from more rigorous scientific language and terminology throughout.

Study Limitations:

7. The cross-sectional design inherently limits causal inference. This limitation should be explicitly acknowledged and its implications for result interpretation discussed.

8. The use of regional classifications as a proxy for air pollution exposure, rather than direct measurements, introduces potential measurement bias. How might this affect the validity of the pollution-related findings?

9. The HRV measurement protocol lacks sufficient detail. Factors such as measurement duration, diurnal timing, and environmental conditions can significantly influence HRV parameters and should be thoroughly described.

Recommendations for Revision:

10. Provide a comprehensive description of the HRV measurement protocol in the methods section, including all relevant parameters and environmental controls.

11. Expand the discussion to address potential mechanisms underlying the observed lack of association between air pollution exposure and HRV. This should include a critical evaluation of the limitations inherent in using regional classifications as exposure proxies.

12. Consider conducting additional statistical analyses to explore potential interaction effects, particularly between CRF and air pollution exposure, which could reveal more nuanced relationships.

Conclusion:

This study contributes valuable data to the field of cardiac autonomic regulation and its relationship with various health-related factors in a large cohort. While the absence of an association between HRV and air pollution exposure is unexpected, the strong correlations with CRF and age provide important confirmatory evidence. With the suggested revisions and an expanded discussion, this manuscript has the potential to make a significant contribution to the literature on cardiac autonomic regulation and environmental health.

Reviewers' comments:

Reviewer's Responses to Questions

**Comments to the Author**

1. Is the manuscript technically sound, and do the data support the conclusions?

Reviewer #1: Partly

2. Has the statistical analysis been performed appropriately and rigorously? 

Reviewer #1: Yes

3. Have the authors made all data underlying the findings in their manuscript fully available?

Reviewer #1: Yes

4. Is the manuscript presented in an intelligible fashion and written in standard English?

Reviewer #1: No

5. Review Comments to the Author

Reviewer #1: Methodological Considerations:

1. What methodology was employed to assess individual physical activity levels? Was self-reporting utilized? How was strength training accounted for in the analysis?

2. Were ambient air pollution levels measured concurrently with HRV assessments? If not, how was temporal variability in air quality addressed?

3. The study design appears to have omitted consideration of physical activity intensity. What was the rationale for this exclusion, and how might it impact the interpretation of results?

4. The broad exclusion criteria may introduce selection bias. What measures were implemented to mitigate this potential confounding factor?

Results Interpretation and Discussion:

5. The discrepancy between this study's findings and previous literature warrants further elucidation. A more comprehensive comparative analysis of methodologies and populations might shed light on these inconsistencies.

6. The manuscript would benefit from more rigorous scientific language and terminology throughout.

Study Limitations:

7. The cross-sectional design inherently limits causal inference. This limitation should be explicitly acknowledged and its implications for result interpretation discussed.

8. The use of regional classifications as a proxy for air pollution exposure, rather than direct measurements, introduces potential measurement bias. How might this affect the validity of the pollution-related findings?

9. The HRV measurement protocol lacks sufficient detail. Factors such as measurement duration, diurnal timing, and environmental conditions can significantly influence HRV parameters and should be thoroughly described.

Recommendations for Revision:

10. Provide a comprehensive description of the HRV measurement protocol in the methods section, including all relevant parameters and environmental controls.

11. Expand the discussion to address potential mechanisms underlying the observed lack of association between air pollution exposure and HRV. This should include a critical evaluation of the limitations inherent in using regional classifications as exposure proxies.

12. Consider conducting additional statistical analyses to explore potential interaction effects, particularly between CRF and air pollution exposure, which could reveal more nuanced relationships.

Conclusion:

This study contributes valuable data to the field of cardiac autonomic regulation and its relationship with various health-related factors in a large cohort. While the absence of an association between HRV and air pollution exposure is unexpected, the strong correlations with CRF and age provide important confirmatory evidence. With the suggested revisions and an expanded discussion, this manuscript has the potential to make a significant contribution to the literature on cardiac autonomic regulation and environmental health.

6. PLOS authors have the option to publish the peer review history of their article (what does this mean?). If published, this will include your full peer review and any attached files.

Reviewer #1: **Yes: **Amirhossein Poopak, MD, MPH

---

## [Author Response · Author response to Decision Letter 0]

28 Nov 2024

Dear reviewers, 

We would like to thank you for your valuable comments, which have significantly enhanced the quality of our article. We have carefully reviewed all your feedback and incorporated it into the manuscript.

Methodological Considerations:

1. What methodology was employed to assess individual physical activity levels? Was self-reporting utilized? How was strength training accounted for in the analysis?

Authors answer: The level of physical activity (used to distinguish “runners” from “inactive participants”) was measured in an online entry screening survey conducted by a professional social science research and marketing company selected through a publicly announced tender. The questions on physical activity were based on standardized questionnaires (i.e. IPAQ, LTEQ), but focused specifically on moderate to vigorous physical activity only. The primary goal was to distinguish between participants who do and do not meet the WHO recommendation on MVPA minutes per week. In addition to questions related to physical activity level, this online screening was customized with questions related to the target group of interest (runners vs. inactive) and included questions on the specifics of running (e.g., distance per week). Questions specifically related to strength training were not part of the screening as strength training was not explicitly included in the definitions of the groups of interest. 

2. Were ambient air pollution levels measured concurrently with HRV assessments? If not, how was temporal variability in air quality addressed?

Authors answer: 

The air pollution assessment in this study was not performed concurrently with the HRV measurements. Instead, we utilized data from the Czech Hydrometeorological Institute spanning 2017 to 2021, the period during which participant data were collected. This study aims to address whether long-term habitation in a region with a long history of elevated benzo[a]pyrene and particulate matter exposure—specifically, the Moravian-Silesian region, identified as one of the most polluted areas in the European Union (as documented by Sram et al. (2013), Hunova et al. (2020), Michalik et al. (2022), and Machaczka et al. (2023))—influences the level of cardiac vagal modulation. We acknowledge the methodological limitations inherent to this approach, including the lack of temporal specificity and the absence of real-time air pollution measurement, and discuss these in the study’s limitations section. Despite these constraints, we believe this study provides valuable insights and unique data drawn from a large cohort regarding chronic exposure.

3. The study design appears to have omitted consideration of physical activity intensity. What was the rationale for this exclusion, and how might it impact the interpretation of results? 

Authors answer: We acknowledge that the intensity of physical activity is another variable that may influence HRV levels, and we have discussed this factor on page 9-10, lines 265–270, along with relevant references. However, we opted to include in our analysis independent variables identified in the literature as contributors to the development of non-communicable diseases, such as cardiorespiratory fitness (CRF), visceral adipose tissue, and ambient air pollution, rather than focusing solely on physical activity intensity. We would also like to thank you for this comment, as it presents an excellent idea for potential analysis in future studies. 

4. The broad exclusion criteria may introduce selection bias. What measures were implemented to mitigate this potential confounding factor?

Authors answer: We stratified the sample by age, gender, location, and physical activity status (active runner vs. inactive). To avoid the effects of various chronic health conditions, we solely focused on subjectively healthy subjects, which made it necessary to apply strict exclusion criteria. We know this approach does not allow general conclusions, although it allows and secures to prove the effects of air pollution and PA more strictly. We addressed this limitation in the limits section.

Results Interpretation and Discussion:

5. The discrepancy between this study's findings and previous literature warrants further elucidation. A more comprehensive comparative analysis of methodologies and populations might shed light on these inconsistencies.

Authors answer: The discrepancy between our findings and those of studies reporting a negative impact of air pollution on heart rate variability (HRV) can be attributed to methodological differences. Notably, our approach of assigning participants to experimental and control groups based exclusively on regional pollution differences is relatively novel in this domain. This methodological distinction may contribute to the variance in findings compared to studies employing direct pollution measurements or quantifying individual exposure levels. Additionally, our study sample, which comprised only healthy individuals aged 18-65 years stratified by sex and age, contrasts with other research that predominantly involves smaller samples, individuals at increased cardiovascular risk or with pre-existing cardiovascular conditions, or older adults. We have also incorporated findings from two studies that examine the effects of physical activity in polluted environments and its potential impact on health, offering this as a possible explanation. We welcome the suggestions and included additional information in the discussion. 

6. The manuscript would benefit from more rigorous scientific language and terminology throughout.

Authors answer: We thank the reviewer for this comment. A native speaker changed the text accordingly.

Study Limitations:

7. The cross-sectional design inherently limits causal inference. This limitation should be explicitly acknowledged and its implications for result interpretation discussed.

Authors answer: We fully acknowledge the inherent limitations of cross-sectional studies, which restrict causal inference. This important consideration is explicitly addressed in the Strengths and Limitations section (p. 11, lines 338-340). We want to clarify that our study does not imply a causal relationship; instead, we have intentionally used the term 'association' to accurately reflect our findings. Specifically, we state: 'The present study shows that cardiorespiratory fitness and age were significantly associated with heart rate variability, independent of living in a high air-polluted region and the amount of visceral adipose tissue in healthy runners and inactive individuals. By emphasizing these associations rather than causation, we aim to provide a more nuanced interpretation of our results while acknowledging the need for further research to explore these relationships in greater depth.

8. The use of regional classifications as a proxy for air pollution exposure, rather than direct measurements, introduces potential measurement bias. How might this affect the validity of the pollution-related findings?

Authors answer: We are fully aware that direct pollution measurements would yield more precise data on pollutant levels. Nevertheless, despite the absence of direct exposure assessments in this study, we suggest that our large sample size offers valuable contributions to the existing literature. Within the study’s limitations, we also recognize that pollution levels can vary across different areas within a region. Based on an analysis of monitoring station data across districts in the Moravian-Silesian region, we identified districts where pollution levels did not exceed recommended thresholds, and individuals residing in these areas were excluded from recruitment. This approach may have partially mitigated the impact of regional pollution variability on our findings.

9. The HRV measurement protocol lacks sufficient detail. Factors such as measurement duration, diurnal timing, and environmental conditions can significantly influence HRV parameters and should be thoroughly described. 

Authors answer: Details regarding the measurement duration and diurnal timing are provided on page 5, lines 157 and 158. We have added specific information indicating that the measurement was conducted immediately upon waking and continued until 7:00 am. Additionally, instructions given to participants by the researcher the evening prior to measurement are outlined on page 5, lines 159 to 161.

Recommendations for Revision:

10. Provide a comprehensive description of the HRV measurement protocol in the methods section, including all relevant parameters and environmental controls.

Authors answer: Thank you very much for this recommendation, we expanded the methods section regarding this point in page 5 from line 159 to line 161.

11. Expand the discussion to address potential mechanisms underlying the observed lack of association between air pollution exposure and HRV. This should include a critical evaluation of the limitations inherent in using regional classifications as exposure proxies.

Authors answer: Thank you very much for this recommendation, we expand the discussion regarding this point in pages 10-11 from line 297 to line 321. Based on this, we added information from two additional studies, which were not part of the initial manuscript, to the discussion section. 

References:

40. Juneja Gandhi, T., Garg, P. R., Kurian, K., Bjurgert, J., Sahariah, S. A., Mehra, S., & Vishwakarma, G. (2022). Outdoor Physical Activity in an Air Polluted Environment and Its Effect on the Cardiovascular System-A Systematic Review. International journal of environmental research and public health, 19(17), 10547. https://doi.org/10.3390/ijerph191710547

41. Andersen, Z. J., de Nazelle, A., Mendez, M. A., Garcia-Aymerich, J., Hertel, O., Tjønneland, A., Overvad, K., Raaschou-Nielsen, O., & Nieuwenhuijsen, M. J. (2015). A study of the combined effects of physical activity and air pollution on mortality in elderly urban residents: the Danish Diet, Cancer, and Health Cohort. Environmental health perspectives, 123(6), 557–563. https://doi.org/10.1289/ehp.1408698

12. Consider conducting additional statistical analyses to explore potential interaction effects, particularly between CRF and air pollution exposure, which could reveal more nuanced relationships.

Authors answer: Thank you very much for your valuable suggestion. Given the nature of our data, however, we believe exploring the suggested interaction may have limited significance in this case. Our analysis was conducted with separate regional data, not with more granular values that would provide specific exposure levels for each respondent. With such detailed data, investigating the interaction with CRF would indeed be meaningful. This is an excellent idea for future research, and we appreciate your insightful recommendation.

---

## [Editor Report · Decision Letter 1]

2 Dec 2024

Association between cardiac autonomic regulation, visceral adipose tissue, cardiorespiratory fitness and ambient air pollution: 4HAIE Study (Program – 4).

PONE-D-24-39037R1

Dear Dr. Dostal,

We’re pleased to inform you that your manuscript has been judged scientifically suitable for publication and will be formally accepted for publication once it meets all outstanding technical requirements.

Kind regards,

Aida Fallahzadeh

Guest Editor

PLOS ONE

Additional Editor Comments (optional):

Dear Dr. Dostal,

We are pleased to inform you that your manuscript, titled “, Association between cardiac autonomic regulation, visceral adipose tissue, cardiorespiratory fitness and ambient air pollution: 4HAIE Study (Program – 4)” has been accepted for publication in [Journal Name].

The revisions you submitted were thoroughly reviewed, and they have significantly strengthened the scientific quality and presentation of your work. We appreciate your effort in addressing the reviewers’ comments and enhancing the clarity and impact of the manuscript.
---

## [Editor Report · Acceptance letter]

11 Dec 2024

PONE-D-24-39037R1 

PLOS ONE

Dear Dr. Dostal, 

I'm pleased to inform you that your manuscript has been deemed suitable for publication in PLOS ONE. Congratulations! Your manuscript is now being handed over to our production team.

Kind regards, 

on behalf of

Dr. Aida Fallahzadeh 

Guest Editor

PLOS ONE